# m6A Modification—Association with Oxidative Stress and Implications on Eye Diseases

**DOI:** 10.3390/antiox12020510

**Published:** 2023-02-17

**Authors:** Yueqi Ni, Hong Zhang, Liang Chu, Yin Zhao

**Affiliations:** 1Department of Ophthalmology, Tongji Hospital, Tongji Medical College, Huazhong University of Science and Technology, Wuhan 430030, China; 2Hepatic Surgery Center, Tongji Hospital, Tongji Medical College, Huazhong University of Science and Technology, Wuhan 430030, China

**Keywords:** oxidative stress, m6A modification, eye diseases, treatment

## Abstract

Oxidative stress (OS) refers to a state of imbalance between oxidation and antioxidation. OS is considered to be an important factor leading to aging and a range of diseases. The eyes are highly oxygen-consuming organs. Due to its continuous exposure to ultraviolet light, the eye is particularly vulnerable to the impact of OS, leading to eye diseases such as corneal disease, cataracts, glaucoma, etc. The N6-methyladenosine (m6A) modification is the most investigated RNA post-transcriptional modification and participates in a variety of cellular biological processes. In this study, we review the role of m6A modification in oxidative stress-induced eye diseases and some therapeutic methods to provide a relatively overall understanding of m6A modification in oxidative stress-related eye diseases.

## 1. Introduction

Oxidative stress is an imbalance between oxidants and antioxidants in favor of the oxidants, leading to a disruption of redox signaling and control and/or molecular damage [1]. Oxidants are particles that have the ability to accept electrons and thus oxidize other molecules. Reactive oxygen species (ROS) are oxidants and are frequently produced in all aerobic organisms, including free radicals such as hydroxyl radical (HO^•^) and superoxide (O_2_^•−^) and non-radicals that are typically less reactive, such as singlet oxygen (^1^O_2_), hydrogen peroxide (H_2_O_2_), and ozone (O_3_). The ROS are mainly derived from the mitochondrial electron transport chain, changes in metal valence, and enzymatic processes (e.g., xanthine oxidase, NADPH oxidase) [2]. In addition, mitochondria are the main source of ROS in mammals’ cells. In mitochondria, ROS are formed as a by-product of anaerobic metabolism when electrons from NADH and FADH_2_ are transported to reduce O_2_ to H_2_O [3].

The ROS are major cellular intermediates, that can undergo redox reactions and form oxidative modifications on biological macromolecules, thus mediating redox signaling and biological functions [4]. In addition, low levels of ROS production are required to maintain physiological functions, including host defense, proliferation, gene expression, and signal transduction [5]. Under physiological conditions, the toxicity of ROS is counteracted by the antioxidant defense system. Enzymes and non-enzymatic antioxidants constitute the body’s complex defense system against oxidative stress. The endogenous antioxidant defense system enzymes are represented by superoxide dismutases (SODs), glutathione S-transferases (GSTs), glutathione peroxidases (GPXs), thioredoxins (TRXs), thioredoxin peroxidases (TRXPs), catalases (CAT), and peroxiredoxins (PRDXs). Nonenzymatic antioxidant molecules include small molecules such as vitamins E and C, carotenoid, and flavonoids [2]. However, when ROS levels exceed the capacity of the antioxidant and repair systems, ROS can compete for the paired electrons of intracellular molecules, causing lipid peroxidation, protein modifications, and chromosomal and mitochondrial DNA (mtDNA) lesions that alter information transmission and gene expression. Abnormal ROS accumulation eventually leads to autophagy, apoptosis, and necrosis, and triggers the dysfunction of tissues and organs, which induces further ROS generation in a vicious cycle [1].

Oxidative stress is believed to be a common pathological feature in various types of eye diseases from, anterior to the posterior segment. Recent studies have demonstrated that oxidative stress is closely related to RNA modification. RNA modification is a post-transcriptional process that changes the chemical composition of RNA and potentially regulates gene expression or stability. More than 170 nucleoside modifications have been identified [6]. It can occur in different RNA species, such as transfer RNA (tRNA), messenger RNA (mRNA), ribosomal RNA (rRNA), microRNA (miRNA), transfer-messenger RNA (tmRNA), long noncoding RNA (lncRNA), small nuclear RNA (snRNA), small nucleolar RNA (snoRNA), and so on [7]. The prevalent mRNA modifications include N6-methyladenosine (m6A), 5-methylcytosine (m5C), 5-hydroxymethylcytosine (hm5C), N1-methyladenosine (m1A), ribose methylation (2′-O-Me), inosine (I), pseudouridine (Ψ), and uridine (U) [8]. Among various types of RNA modification, m6A is the most well-studied.

In this review, we describe the role of RNA m6A modification in the development of oxidative stress-related eye diseases and other common diseases and the potential treatments for eye diseases based on m6A modification.

## 2. m6A Modification

The M6A modification refers to the methylation of the sixth nitrogen atom of adenylate and has been widely confirmed in viruses, yeast, plants, insects, and mammals [9,10]. They are most frequently found in RRACH motifs (R = A/G, H = A/C/U) [11]. m6A modification is not equally distributed across the transcriptome, and is enriched in a subset of consensus sequences near stop codons and 3′UTRs [12]. m6A is involved in almost all RNA metabolism processes, including processing, splicing, translation, exporting, and degradation [13]. Thus, dynamic m6A modification is important for many physiological and pathological processes. The RNA m6A modification process is reversibly and dynamically regulated by three types of enzymes: m6A methyltransferase (“writers”), m6A demethylase (“erasers”), and m6A-binding protein (“readers”).

### 2.1. Writers

The writers consist of a complex including methyltransferase-like 3 (METTL3), METTL14, METTL16, zinc finger CCCH-type containing 13 (ZC3H13), Wilms tumor 1 associated protein (WTAP), RNA-binding motif protein 15 (RBM15), and vir-like m6A methyltransferase-associated protein (VIRMA) [14]. The m6A modification is mainly catalyzed by the core methylase complex METTL3-WTAP-METTL14. METTL3 is the predominant catalytic subunit of the writer complex that transfers a methyl group from S-adenosylmethionine (SAM) to an adenosine in RNA [15]. METTL3 alone merely exhibits weak activity, while METTL14 stabilizes the METTL3 conformation as an RNA-binding platform to increase METTL3 catalytic activity [16]. WTAP lacks a catalytic methylation domain and therefore has no methyltransferase activity [17]. It can interact with the METTL3-METTL14 heterodimer, thereby facilitating m6A deposition [18]. 

### 2.2. Erasers

The m6A methylation can be reversed via m6A demethylases AlkB homolog 5 (ALKBH5), ALKBH3, or fat mass and obesity-associated protein (FTO). They all belong to the alpha-ketoglutarate-dependent dioxygenase family [19]. FTO oxidizes m6A to the intermediate N6-hydroxymethyl adenosine (hm6A), which can be further oxidized to form N6-formyladenosine (f6A). hm6A and f6A naturally decompose into adenine (A) in aqueous solution [20]. Unlike the oxidative demethylation of FTO, ALKBH5 works by directly removing the methyl group from the methylated adenosine of m6A [21]. 

### 2.3. Readers

The readers mainly consist of IGF2 mRNA binding proteins (IGF2BP1/2/3), the YTH domain protein family (YTHDC1/2, YTHDF1/2/3), eukaryotic initiation factor 3 (eIF3), and the heterokaryotic nuclear RNA protein family (HNRNPC, HNRNPG). The YTH domain can selectively bind to the m6A site in RNA. YTHDC1 promotes the nuclear export of mRNA and regulates mRNA splicing [22,23]. YTHDC2 enhances translation efficiency while decreasing the abundance of its target mRNAs [24]. YTHDF1 and YTHDF3 promote the translation of m6A-modified mRNA. However, YTHDF2 degrades m6A-modified transcripts by recruiting the CCR4-NOT deadenylase complex [25]. IGF2BPs contribute to the stability and translation of target mRNAs through recruiting RNA stabilizers [26]. HNRNPC and HNRNPG participate in the processing of pre-mRNA and affect its stability, splicing, export, and translation [27]. The eIF3 is an eukaryotic initiation factor, while it also has the function of promoting the translation of mRNAs by binding to the m6A sites at the 5′UTR [28]. 

## 3. m6A Modification and Oxidative Stress-Related Eye Diseases

The eyes are highly exposed to light and metabolically active, making them particularly vulnerable to oxidative damage [29]. Oxidative stress can damage eye tissues, leading to changes in tissue structure and function, vascular abnormalities, glial dysfunction, apoptosis, and so on, which can cause pathological changes of the cornea, anterior chamber, lens, uvea, and retina [30]. Moreover, it has been found that m6A modification plays a part in the occurrence and development of many oxidative stress-related eye diseases (Figure 1, Table 1).

### 3.1. Corneal Disease

The cornea acts as an initial physical and biochemical barrier to protect the inner eye tissue from external environmental damage, and is directly exposed to the atmosphere and high levels of oxygen, so strong antioxidant mechanisms are needed to combat oxidative stress [31]. The tear film and cornea contain a variety of enzymatic and non-enzymatic antioxidants, especially glutathione (GSH) and vitamin C [32]. The exogenous sources of ROS in the cornea include environmental factors, exposure to UV radiation, and infection, whereas the endogenous sources mainly come from the highly metabolically active sodium-potassium ATPase complex [33]. Oxidative stress caused by excessive production of ROS has been considered an important factor in ocular surface diseases [34,35]. 

Keratitis is characterized by tears, visual loss, pain, eye irritation, changes in corneal integrity, and corneal neovascularization [36]. The m6A level of corneal tissue increased after fungal infection, which might be the result of the upregulated expression of METTL3 and METTL14. Further studies established m6A modification profiles in *F. solani*-induced corneal keratitis and found that m6A methylation might regulate a variety of significant signal transduction pathways in fungal keratitis, such as the PI3K-Akt signaling pathway [37]. 

Corneal neovascularization occurs due to a number of ocular damages, including trauma, inflammation, infection, degeneration, ischemia, and loss of the limbal stem cell barrier [38]. FTO silencing could increase the m6A level in proangiogenic genes such as focal adhesion kinase (*Fak*), resulting in decreased RNA stability to promote RNA decay through YTHDF2, thus affecting angiogenesis and endothelial cell function during corneal neovascularization [39]. Another study constructed endothelial-specific METTL3 knockout mice, showing that knockout of METTL3 inhibited corneal neovascularization [40]. Moreover, limbal stem cell-specific METTL3 knockout mice presented faster corneal injury repair and less neovascularization by targeting the AHNAK nucleoprotein (*Ahnak*) and DNA damage inducible transcript 4 (*Ddit4*), which were involved in stem cell differentiation, maintenance, and proliferation [41]. 

### 3.2. Cataract

The lens is a transparent tissue with complex optical properties that transmits and focuses light on the retina. Cataract is caused by the loss of transparency of the lens due to its opacity, which is a direct result of oxidative stress [42]. When young, the lens is usually protected from oxidative damage by a powerful system of oxygen radical scavengers that use GSH as the main antioxidant to detoxify ROS [43]. However, protective systems weaken with age, and long-term exposure to oxidative stress can cause cumulative damage to lens cells. Case-control studies have shown that levels of antioxidant enzymes (SOD, GPx, CAT) in cataract patients’ serum were decreased, while levels of oxidative stress products (malondialdehyde MDA, 8-OH 2-deoxyguanosine 8-OHdG, advanced oxidation protein product AOPP, 4-hydroxy-2-nonenal 4-HNE, etc.) were increased [44,45]. In lenses, cellular redox status imbalances induce increased production of mitochondrial ROS; therefore, oxidation of proteins, lipids, and DNA can be found in cataract lenses [46,47]. The aged crystal lens gradually becomes opaque and yellow, exhibiting blue light-filtering characteristics [48]. It has been suggested that intraocular lens implantation may cause retinal dysfunction due to the higher transmittance of short-wavelength visible light compared with that of aged lenses [49]. However, based on the current best available clinical evidence, the effect of blue light-filtering IOLs in cataract surgery on retinal health and function remains unclear [50,51,52]. 

A recent study performed genome-wide profiling of m6A-tagged circRNAs between controls and age-related cortical cataract (ARCC). This study found a dynamic characteristic of m6A modification in lens epithelial cells that was associated with ARCC development [53]. 

In addition, diabetic cataract was also associated with abnormal m6A modifications. METTL3 was upregulated in diabetic anterior lens capsule and high glucose-treated human lens epithelial cells. METTL3 targeted the 3′UTR of intercellular adhesion molecule 1 (*ICAM-1*) to stabilize mRNA stability, which mediated apoptosis of human lens epithelial cells [54]. 

### 3.3. Glaucoma

Glaucoma is a degenerative disease that affects the anterior and posterior segments of the eye and is characterized by cupping of the optic disc, loss of retinal ganglion cells (RGCs), and thinning of the retinal nerve fiber layer [55]. Increased intraocular pressure (IOP), age, and a genetic background are the leading risk factors for glaucoma [56]. 

Poor outflow of aqueous humor (AH) can lead to high IOP, and trabecular meshwork (TM) is the key component of the AH outflow pathway. The trabecular meshwork is the most susceptible to oxidative stress injury in the anterior chamber because of its location, stress sensitivity, lack of blood supply, and constant exposure to ultraviolet radiation and byproducts of cell metabolism [57]. Oxidative stress induces TM cell damage, such as activation of inflammatory pathways, cell autophagy and apoptosis, remodeling of cytoskeletal structure, extracellular matrix accumulation, and mitochondrial damage [58,59,60,61,62]. To combat oxidative stress, TM cells express a number of redox agents, including SOD, glutathione reductase, and peroxidase [63]. 

Not only do ROS cause damage to the TM, but subsequent inflammation and large amounts of ROS production also cause oxidative injury to the retina and optic nerve, which are characteristic damage sites in glaucoma [29]. The evidence supporting the relationship between oxidative stress and glaucoma comes from animal and human studies that have evaluated markers of ROS production, antioxidant levels, and macromolecular oxidative damage under glaucoma conditions [64]. In glaucomatous patients, vision impairment is determined by the degree of RGC loss. It has been shown that different methods of inducing high IOP increased retinal ROS levels, resulting in the degradation of RGCs, while reduction of ROS generation protected RGCs from apoptosis [65,66]. 

One study used a *Six3*-cre mouse line, which is a widely used line to generate retina-specific knockouts, to conditionally knockout YTHDF2 in the retina, demonstrating that YTHDF2 regulated dendrite branching of RGCs and that YTHDF2 deficiency was more resistant to RGC damage from acute ocular hypertension by targeting heat shock protein family A member 12A (*Hspa12a*) and the immunoglobulin superfamily containing leucine rich repeat 2 (*Islr2*) [67]. 

### 3.4. Uveitis

Uveitis is defined as inflammation of the uvea, including the iris, ciliary body, and choroid [68]. In most cases, uveitis is a sporadic disease with unknown etiology, but the eventual loss of vision is always attributed to damage to eye tissue caused by amplification of the inflammatory processes [69]. It is known that proinflammatory cytokines and chemokines induce mitochondrial ROS production. Many studies have shown that oxidative stress products, such as oxidized phospholipids, are present in eye tissues during experimental uveitis, and oxidative damage to mitochondrial DNA occurs in the early stage [70,71]. A study using ROS-deficient (*NCF1-/-*) mice found that, consistent with antioxidant treatment, inhibition of ROS reduced experimental autoimmune uveitis severity in mice [72]. 

Innate immune cells, including microglia, are known to play a crucial role in the pathogenesis of uveitis [73]. YTHDC1 regulated microglia activation by affecting *Sirt1* mRNA stability, and YTHDF1 silencing induced microglia M1 polarization and exacerbated the inflammatory response [74]. 

The retinal pigment epithelium (RPE) is one of the affected cells in uveitis. Decreased FTO expression in RPE cells was found in mice with uveitis. FTO knockdown affected the RNA methylation levels of *ATF4* via both a m6A-dependent manner and the PERK pathway, thereby inhibiting *ATF4* translation. Down-regulation of ATF4 aggravates the damage to tight junctions of RPE by activating the phospho-signal transducer and activator of transcription 3 (p-STAT3) [75]. 

### 3.5. Retinopathy

The retina is one of the most oxygen and energy demanding tissues in the body to generate and transmit visual evoked potential signals and is directly exposed to visible light (400–760 nm) [76]. Energy metabolism naturally generates oxidative stress, and a large amount of ROS is generated [77]. The retina is a highly vulnerable area to oxidative stress, partly because of the high production of ROS, and its photoreceptors are rich in polyunsaturated fatty acids, which are a major target for lipid peroxidation [78]. m6A has an important role in maintaining normal dopaminergic function in the brain. METTL14 depletion impairs striatal learning and alters dopamine signaling [79]. Since the retina is part of the brain, it is reasonable to hypothesize that this impacts eye diseases as well [80]. Recent studies have shown that m6A plays an important role in regulating gene expression or pathways involved in oxidative stress-related retinopathy, especially in age-related macular degeneration.

#### 3.5.1. Diabetic Retinopathy (DR)

The DR is a progressive, asymptomatic diabetic microvascular complication that can cause irreversible retinal damage [81]. Hyperglycemia induces structural changes in blood vessels through ischemia or hyperosmolar injury, therefore leading to an impaired supply of oxygen and nutrients to neurons [82]. 

Hyperglycemia causes various metabolic disorders, such as increased polyol pathway activities, intracellular formation of advanced glycation end products (AGEs), activation of the protein kinase C (PKC) pathway, and the hexosamine pathway [83,84]. In addition, hyperglycemia-induced oxidative stress can also lead to mitochondrial defects, apoptosis, inflammation, lipid peroxidation, and neurodegeneration in the retina [85,86,87,88,89,90]. 

Diabetic stress resulted in an increased level of m6A modification and METTL3 in diabetic retinal vessels and pericytes. Inhibition of METTL3 reduced *PKC-η*, FAT atypical cadherin 4 (*FAT4*), and platelet derived growth factor receptor alpha (*PDGFRA*) mRNA degradation in a YTHDF2-dependent manner, thereby alleviating pericyte dysfunction and retinal vascular complications [91]. Lysine acetyltransferase 1 (KAT1), a histone acetyltransferase, had lower expression in diabetic mouse retinas than in the control groups. In vivo and in vitro experiments found that overexpression of KAT1 induced YTHDF2 to promote the instability of integrin subunit beta 1 (*Itgb1*) mRNA, which in turn attenuates Müller cell activation and neovascularization by inhibiting the FAK/PI3K/AKT signaling pathway [92]. 

Retinal inflammation caused by dysregulated polarization of microglia is considered one of the key pathogenesis of DR. High glucose inhibited the expression of ALKBH5 in microglia, leading to higher levels of m6A modification in *A20* mRNA, which accelerated its degradation and then caused a decreased protein level. The microglia with lower A20 expression tended to polarize into the M1 inflammatory type rather than the M2 anti-inflammatory type, ultimately promoting the occurrence and development of DR [93]. 

The m6A modification can participate in DR-induced RPE injury. A recent study reported that *miR-25-3p* and METTL3 levels in diabetic patients’ peripheral venous blood were lower than those of healthy controls, and high glucose could decrease the expression of *miR-25-3p* and METTL3 in RPE cells. The overexpression of the METTL3-regulated PTEN/AKT signaling pathway by upregulating *miR-25-3p* through the microprocessor protein DGCR8 alleviated the injury of high glucose on RPE cells [94]. Increased *NLRP3* mRNA stability and protein expression in RPE cells were mediated by FTO, thus promoting RPE cell pyroptosis. *miR-192* could reverse the effect of HG on RPE cell pyroptosis by negatively regulating FTO expression [95]. In addition, *circFAT1* promoted autophagy and inhibited pyroptosis in HG-stimulated RPE cells, possibly by binding to YTHDF2 [96]. 

#### 3.5.2. Retinopathy of Prematurity (ROP)

The ROP can be regarded as a pathological compensatory mechanism of abnormal retinal vessels resulting from the arrest of normal retinal neurons and vessels in prematurity. The development of ROP has two distinct phases: Phase-I is characterized by the inhibition of retinal angiogenesis by a hyperoxic environment compared to that in the uterus. In Phase-II, the hypoxic retina upregulates the expression of regulatory substances such as vascular endothelial growth factor (VEGF) and erythropoietin (EPO) to stimulate retinal neovascularization [97]. 

Hypoxia is the inducer of ROS production [98]. During the perinatal period, especially at birth, the newborn is exposed to oxidative stress due to a rapid change from a very low oxygen intrauterine environment to an environment with higher oxygen levels, coupled with an incomplete antioxidant protection system [99]. 

Oxygen-induced retinopathy (OIR) is a model for ROP [100]. In OIR mice, oxidative stress was demonstrated in the retina and induced upregulation of pyroptosis and inhibition of autophagy [101]. 

Hypoxic stress induced elevated levels of m6A modification and METTL3 in human umbilical vein endothelial cells (HUVECs) and mouse retinas. Knockdown or conditional knockout of METTL3 inhibited the angiogenesis of HUVECs and pathological angiogenesis of the cornea and retina, which were the results of oxidative stress. These were achieved by reducing disheveled segment polarity protein 1 (*DVL1*) and LDL receptor related protein 6 (*LRP6*) in a YTHDF1-dependent manner to regulate Wnt signal activation [40]. 

#### 3.5.3. Retinitis Pigmentosa (RP)

The RP is a set of inherited retinopathies in which a mutation causes the death of retinal photoreceptors [102]. In most cases of RP, the degeneration of rod cells occurs first [103]. Rods make up 95% of the cells in the retina’s outer nuclear layer. They are rich in mitochondria and highly metabolically active. Once the rods die, retinal oxygen consumption decreases, leading to an elevated oxygen concentration. This may increase oxidative stress in the retina and enhance oxidative damage to cones, which are responsible for the main human vision and daytime vision [104]. Their antioxidant defenses gradually get overwhelmed, leading to cell damage.

METTL14 was downregulated in lymphoblastoid samples from RP patients, and knockdown of METTL14 inhibited RPE cell phagocytosis and proliferation and promoted RPE cells’ apoptosis. Microtubule-associated protein (MAP2) directly interacts with the pathogenic gene neuronal differentiation 1 (NEUROD1) in RPE cells. METTL14 could bind to *MAP2* mRNA, inhibiting *MAP2* expression through YTHDF2 [105]. 

#### 3.5.4. Age-Related Macular Degeneration (AMD)

AMD is characterized by the loss of photoreceptors and RPE cells in the macula [106]. AMD is associated with several environmental and genetic risk factors, such as aging, smoking, excess sun exposure, and so on, which are linked to increased oxidative stress [107]. Numerous in vitro and in vivo studies have demonstrated that oxidative stress is a major factor affecting AMD pathophysiology [108]. 

RPE is the major site of pathological alterations in AMD. In RPE cells treated with oxidative toxic agents H_2_O_2_, NaIO_3_, or pro-inflammatory cytokine TNF-α, as well as in macular RPE and blood samples from patients with dry AMD, the level of hsa_circ_0000745 (circSPECC1) was reduced. circSPECC1 knockdown induces oxidative damage and irregular lipid metabolism in vitro. In addition, circSPECC1 insufficiency causes structural abnormalities in the mouse RPE and leads to decreased vision and an atrophic fundus. Mechanically, reducing m6A levels in the circSPECC1 transcript can interfere with its nuclear export, with YTHDC1 acting as a potential reader [109]. 

Another research team found that demethylase FTO expression in RPE was upregulated in the Aβ 1-40-induced model of RPE degeneration. However, the specific inhibition of FTO in vivo exacerbated the retinal hypopigmentation and RPE structural disorder induced by Aβ treatment. The protective effect of FTO is achieved by inhibiting PKA/CREB signaling, thereby partially rescuing RPE from degeneration [110]. 

### 3.6. Traumatic Optic Neuropathy (TON)

TON refers to an indirect or direct optic nerve injury, mainly because of head trauma, resulting in a partial or complete loss of optic nerve function [111]. Blunt injury to the optic nerve can cause ROS accumulation in the retina, resulting in increased production of superoxide, decreased superoxide dismutase activity, and activation of the inflammasome [112]. Researchers found the expression of *Fto*, *Alkbh5*, *Mettl3*, and *Wtap* mRNA in the retina was all upregulated after TON, and then they identified genes with altered expression and methylation by combining the analysis of RNA-seq and MeRIP-seq. A total of 689 m6A peaks were differentially downregulated and 2810 m6A peaks were upregulated in TON retinas compared with controls. These altered m6A peaks chiefly correlated with nervous system development, and were associated with the NF-κB signaling pathway, the MAPK signaling pathway, and the TNF signaling pathway [113]. 

### 3.7. Ocular Tumor

Ocular melanoma is the second most common type of melanoma after cutaneous melanoma. It can be divided into conjunctival melanoma (CM) and uveal melanoma (UM) [114,115]. The ROS level of melanoma is significantly higher than that of other types of tumors, which is mainly due to two reasons: exposure to ultraviolet radiation and the synthesis of melanin [116]. Melanin biosynthesis needs oxidation reactions, which involve ROS production that subjects melanocytes to oxidative stress [117]. 

CM and UM clinical human samples showed decreased m6A modification, with lower levels of METTL3 and higher levels of ALKBH5, indicating a poor prognosis. Histidine triad nucleotide binding protein 2 (*HINT2*), proved to be a tumor suppressor gene in ocular melanoma, was then determined to be the m6A-mediated target via MeRIP-Seq and RNA-seq assays. In addition, YTHDF1 promoted the translation of *HINT2* mRNA in a m6A-dependent manner [118]. High levels of ALKBH5 induced proliferation, migration, and invasion of UM cells in vivo and in vitro, and promoted epithelial-mesenchymal transition of UM cell lines C918 and MuM-2B by upregulating Forkhead box M1 (FOXM1) expression [119]. 

Another study used UM clinical samples and other cell lines to reach a different conclusion. The study found that m6A modification was increased in UM cell lines and clinical samples, and the hypermethylation level was beneficial to tumor growth and invasion. Upregulated METTL3 contributed to the progression of UM by targeting *c-Met* to participate in the regulation of cell cycle-related proteins and the AKT signaling pathway [120]. METTL3-mediated m6A was involved in the high expression of beta-secretase 2 (*BACE2*), which had been shown to play an oncogenic role in ocular melanoma [121]. 

The levels of lactic acid and H3K18LA were increased in ocular melanoma. H3K18LA activated the transcription of YTHDF2 by promoting the tumor suppressor gene *PER1* and *TP53* degradation, which accelerated tumorigenesis. This discovery linked histone modifications to RNA modifications [122]. 

Retinoblastoma (RB) is the most common ocular malignancy in children and is highly aggressive. It is caused by a mutated *RB1* gene, which plays a key role in cell cycle regulation [123]. It has been shown that RB cells have higher mitochondrial activity than normal retinal cells. In addition, tigecycline, atovaquone, and salinomycin, which are some novel types of anti-cancer drugs, can inhibit retinoblastoma through mitochondrial dysfunction-dependent oxidative damage [124,125,126]. 

In retinoblastoma cell lines, METTL3 was higher than that in normal ARPE-19 cells, and upregulation of METTL3 promoted the growth of retinoblastoma via the PI3K/AKT/mTOR signaling pathway [127]. 

Here is a summary for m6A modification in the oxidative stress-related eye diseases (Table 1).

**Table 1 antioxidants-12-00510-t001:** The functions of m6A regulatory enzymes in oxidative stress-related eye diseases.

Eye Diseases	Total m6A Level	Enzymes	Functions
Corneal Neovascularization	N/A	METTL3	Targets stem cell regulatory factors *Ahnak* and *Ddit4* to regulate limbal stem cell proliferation and migration [41].
	Down	FTO	Increases the expression of FAK to promote vascular endothelial cell function and angiogenesis [39].
Cataract	Up	METTL3	Regulates the lens epithelial cells proliferation and apoptosis by targeting the 3’UTR of *ICAM-1* to stabilize mRNA stability [54].
Glaucoma	N/A	YTHDF2	Targets *Hspa12a* and *Islr2* to affect the tolerance of RGC to acute intraocular hypertension [67].
Uveitis	Down	YTHDF1	Increases *Sirt1* expression thus restraining M1 polarization and migration in microglia [74].
	N/A	FTO	Promotes *ATF4* expression via both m6A-dependent manner and PERK pathway, thereby reducing the inflammatory response [75].
Diabetic Retinopathy	N/A	METTL3	Targets *miR-25-3p*/PTEN/AKT signaling cascade to enhance RPE cell viability [93].
	Up	METTL3	Reduces *PKC-η*, *FAT4* and *PDGFRA* mRNA expression thus aggravating pericyte dysfunction [90].
	N/A	FTO	Promotes the mRNA stability of *NLRP3* to activate pyroptosis in RPE cells [94].
	N/A	ALKBH5	Protects *A20* mRNA from degradation to reduce M1 microglia polarization [92].
	N/A	YTHDF2	Promotes the instability of *IGTB1* mRNA which is a positive regulator of the FAK/PI3K/AKT signaling pathway [91].
Retinopathy of Prematurity	Up	METTL3	Regulates Wnt signaling activation by targeting *LRP6* and *DVL1* [40].
Retinitis Pigmentosa	Down	METTL14	Inhibits expression of MAP2 which interacts with NEUROD1 to induce pathologic changes in RPE cells [104].
Age-related Macular Degeneration	N/A	FTO	Rescues RPE from degeneration by inhibiting PKA/CREB signaling [109].
Ocular Melanoma	Down	METTL3, ALKBH5	Regulates the translation of the tumor suppressor gene *HINT2* to regulate ocular melanoma tumorigenesis [117].
	N/A	ALKBH5	Increases FOXM1 expression to promote invasion, migration, and epithelial–mesenchymal transition in UM cells [118].
	Up	METTL3	Enhances UM cell proliferation, migration, and invasion by promoting *c-Met* translation [119].
	N/A	METTL3	Increases RNA and protein level of BACE2, which accelerates tumorigenesis via TMEM38B/Ca^2+^ pathway [120].
	N/A	YTHDF2	Promotes degradation of *PER1* and *TP53* mRNA, thereby accelerating tumorigenesis of ocular melanoma [121].
Retinoblastoma	N/A	METTL3	Promotes RB progression in vivo and in vitro through PI3K/AKT/mTOR pathway [126].

## 4. Treatment

The eye is a very small organ and a closed environment, which makes it easy to get close to the target by suprachoroidal, intravitreal, or subretinal injection, and the optimal concentration can be maintained for a long time by injecting only a small amount of reagent. The blood-retinal barrier ensures the immune privilege of the eye, which prevents foreign antigens or viruses from being rejected by the immune system, and reduces the risk of systemic transmission of therapy vectors [128]. 

The main delivery systems for gene therapy in the eyes include viral and nonviral vectors. Adenovirus is the most widely studied virus, and adeno-associated virus (AAV) has gained popularity because of its better safety profile [129]. The serotypes determine the cell and tissue specificity of the AAV. Nanoparticles (NPs), with dimensions between 1 and 100 nanometers, can carry a variety of genetic material, including DNA, RNA, and ribonucleoprotein, due to their large capacity [130]. NPs have the ability to reduce toxicity, increase hydrophobic drug solubility, prolong drug retention time, sustain drug release, and enhance drug penetration across the eye barrier [131]. 

The CRISPR/Cas system is an innate immune system from bacteria and archaea, consisting of CRISPR sequences and highly diverse Cas proteins [132]. CRISPR/Cas9 can achieve silence/knockout, knockin/replacement, point mutation, and other operations on specific genes. It can also be used for genome-wide screening of targets for therapeutic purposes [133]. 

A recent study fused CRISPR-dead Cas9 (dCas9), which lacks the ability to cut DNA, with a single chain m6A methyltransferase or full length FTO/ALKBH5 to achieve sgRNA-guided site-specific m6A modification in mouse embryonic fibroblasts (MEF) and HeLa cells [134]. Many studies have turned to CRISPR/Cas13-based platforms. A research team successfully manipulated a single m6A site in RNA by binding catalytically inactive CasRx (dCasRx) to METTL3 or ALKBH5, using HEK293T, glioma stem cells (GSC) 468, and GSC 3565 cells [135]. Another team achieved the fusion of dPspCas13b and ALKBH5 in HEK293T and HeLa cells. This fusion protein is called dm^6^ACRISPR, which specifically demethylates m6A of targeted mRNA [136]. Researchers also fused catalytically inactive PspCas13b protein with YTHDF1 and YTHDF2 in HEK293T and Hela cells and proved that it could implement the original functions of these two kinds of m6A readers: YTHDF2 induces degradation, and YTHDF1 enhances translation [137]. 

The CRISPR/Cas system is already used in clinical trials mainly to treat tumors, immune system disorders, and inherited anemia. (ClinicalTrials.gov Identifier: NCT04774536, NCT03057912, NCT03164135, NCT04426669, NCT03545815, NCT04637763, NCT04037566, NCT05477563, etc.) It has also been applied in the field of ophthalmology. BDgene Company completes Phase-I/II clinical trials using a novel CRISPR/Cas9 mRNA instantaneous gene editing product named “BD111” in refractory herpetic viral keratitis (NCT04560790). Researchers evaluate the dosing and safety of BD111 and find that it only needs to be injected once and reduces the risk of an immune response and off-target gene editing. BD111 received FDA approval as an orphan drug on June 24, 2022. Another ongoing trial from Editas Medicine Incorporated uses a single escalating dose of “EDIT-101,” which is AAV5 with DNA encoding two gRNAs and SaCas9, administered via subretinal injection in participants with Leber Congenital Amaurosis 10 (LCA10) (NCT03872479). To date, no dose-limiting toxicities or serious adverse events have been reported. The early data are promising. The mid-dose cohort shows improved vision in the Berkeley rudimentary vision test and the ORA visual navigation course.

## 5. Conclusions

Artificial intelligence assisted technology has been widely used in drug candidate discovery and development. Structure-based virtual screening of compounds is an important approach to obtaining effective molecular drugs, and this technique can also be used in the field of ophthalmology [138,139]. At present, some small-molecule activators or inhibitors targeting m6A modification enzymes have been reported. Most of them are still in the pre-clinical stage. A small-molecule ligand was experimentally qualified as a METTL3-14-WTAP activator by binding to the active site of the METTL3-14-WTAP complex [140]. The development of m6A regulatory enzymes’ inhibitors has recently been the focus of therapy (Table 2).

Currently, several biotechnology companies have announced that they have developed small-molecule inhibitors of the METLL3-METTL14 complex targeting different types of cancer, such as AML and non-small cell lung cancer (NSCLC), and they are preparing to conduct clinical phase-I trials in 2021–2022. Other targets, such as eraser FTO and reader YTHDF family, are also potential targets for attention [151]. 

The drugs mentioned above are mainly for cancer. Given that there is a close relationship between eye diseases and m6A modification and many applications of gene therapy in eye diseases, it is reasonable to speculate that m6A modification in eye diseases will become a promising therapeutic target in the future.

## Figures and Tables

**Figure 1 antioxidants-12-00510-f001:**
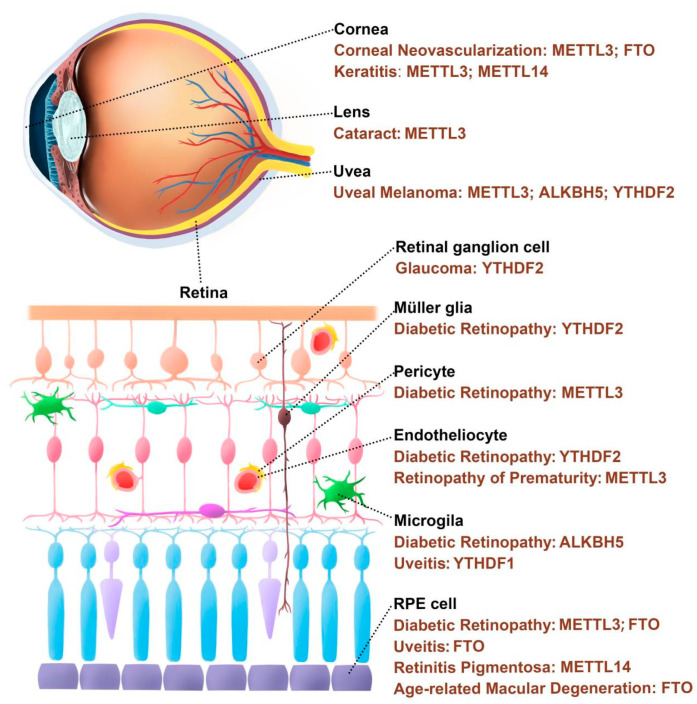
m6A regulatory proteins are involved in the occurrence and development of eye diseases.

**Table 2 antioxidants-12-00510-t002:** Inhibitors targeting m6A regulatory enzymes and their functions.

Targets	Inhibitors	Diseases or Cell Lines	Functions
FTO	Rhein	BE(2)-C cells	Reversibly binds FTO enzyme and competitively prevents the recognition of m6A substrates [141].
	Meclofenamic Acid	HeLa cells	Inhibits FTO demethylation of an m6A-containing ssDNA or ssRNA [142].
	R-2HG	human leukemia cell lines and leukemic mouse models	Inhibits leukemia cell proliferation/viability and promotes cell-cycle arrest and apoptosis [143].
	FB23/FB23-2	human acute myeloid leukemia (AML) cell lines and primary blast AML cells	Suppresses proliferation and promotes the differentiation/apoptosis of human AML cell line cells and primary blast AML cells in vivo and in vitro [144].
	Fluorescein	HeLa cells	Inhibits FTO demethylation inside live cells [145].
	CS1/CS2	AML cell lines, primary AML cells, and leukemic mouse models	Attenuates leukemia stem/initiating cell self-renewal, reprogram immune response, and immune evasion [146].
	Dac51	B16-OVA, LLC, and MC38 cells, and tumor-bearing mice	Blocks FTO-mediated immune evasion [147].
ALKBH5	MV1035	U87-MG and H460 cells	Reduces U87 glioblastoma cell line migration and invasiveness [148].
METTL3	STM2457	AML cell lines, and leukemic mouse models	Reduces AML growth and increases differentiation and apoptosis [149].
	UZH1a/UZH1b	MOLM-13, U2OS, and HEK293T cells	Reduces the m6A/A ratio in mRNAs of AML MOLM-13 cells, osteosarcoma U2OS cells, and the embryonic kidney cell line HEK293T [150].

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
