# Peer review of "m6A Modification—Association with Oxidative Stress and Implications on Eye Diseases"

_antioxidants, 2023, doi:10.3390/antiox12020510_

Round 1

Reviewer 1 Report

The authors review the role of m6A modifications and some therapeutic strategies in oxidative stress-induced diseases, especially eye diseases.

This manuscript is well written.

If the Editor makes some improvements before publication, the value of this paper would increase.

Major Points:

1. From P.11 to P.24, there are descriptions other than ophthalmology about m6a modification. Important statement, but it doesn't match the title.

This manuscript is huge. In line with this title, it would be better to make a separate paper for descriptions other than ophthalmology. By doing so, there would be a focus on the ophthalmology segment, which would increase this current value.

2. Line 35

“ROS are currently considered to be important signal transduction molecules or regulators in biological systems, rather than merely harmful by-products of metabolism.” 

It was reported 30 years ago that small amounts of ROS act as second messengers.

Therefore, it is better to change this description.

3. Line 163 - 165

Although an artificial lens with a blue light filter is implanted in cataract surgery, UV-mediated retinal damage may be exacerbated by the loss of the natural protection of the lens.47  

If there are any reports of retinal damage caused by UVA passing through the IOL in clinical cases, please show them. In vitro papers are cited in these reviews. If the IOL cannot protect against blue light and long-term irradiation causes retinal damage, the reviewer thinks this is a very important report. This is because many countries use IOL lenses that cut blue light for no reason.

4. Line 225 - 227

“The retina is one of the most oxygen and energy demanding tissues in the body to generate and transmit visual evoked potential signals, and is directly exposed to visible light and ultraviolet radiation, thus, a large amount of ROS are generated.71 

The three types of ultraviolet rays that reach the ground affect the anterior segment of the eye, but are mostly absorbed by the cornea and lens and do not reach the retina.

Numerous in vitro experiments have investigated UV damage to RPE.

However, it is not clear whether this experimental system affects the retinal diseases seen in actual clinical practice.

I'm not sure Ref.71 is the best citation.

Author Response

Comment 1. From P.11 to P.24, there are descriptions other than ophthalmology about m6a modification. Important statement, but it doesn't match the title.

This manuscript is huge. In line with this title, it would be better to make a separate paper for descriptions other than ophthalmology. By doing so, there would be a focus on the ophthalmology segment, which would increase this current value.

Response 1: Thanks for the referee’ s suggestion. Review 3 also pointed out that section 5 was unnecessary. We therefore deleted section 5, figure2, table2, associated abbreviations, and original references 124-152.

Comment 2. Line 35

“ROS are currently considered to be important signal transduction molecules or regulators in biological systems, rather than merely harmful by-products of metabolism.”

It was reported 30 years ago that small amounts of ROS act as second messengers.

Therefore, it is better to change this description.

Response 2: Thanks for the referee’ s suggestion. This description has been changed in revised manuscript Line 33 as “ROS are major cellular intermediates, which can undergo redox reactions and form oxidative modifications on biological macromolecules, thus mediating redox signalling and biological functions.”

Comment 3. Line 163 - 165

Although an artificial lens with a blue light filter is implanted in cataract surgery, UV-mediated retinal damage may be exacerbated by the loss of the natural protection of the lens.47 

If there are any reports of retinal damage caused by UVA passing through the IOL in clinical cases, please show them. In vitro papers are cited in these reviews. If the IOL cannot protect against blue light and long-term irradiation causes retinal damage, the reviewer thinks this is a very important report. This is because many countries use IOL lenses that cut blue light for no reason.

Response 3: Thanks for the referee’ s suggestion. This sentence is not accurate, and we have changed it to “The aged crystal lens gradually becomes opaque and yellow, exhibiting blue light-filtering characteristics. It has been suggested that intraocular lens implantation may cause retinal dysfunction due to the higher transmittance of short-wavelength visible light compared with that of aged lens. However, based on the current best available clinical evidence, the effect of blue light-filtering IOLs in cataract surgery on retinal health and function remains unclear.” The passage is in the revised manuscript line 161-167. We have added new references as Ref.48-52.

Comment 4. Line 225 - 227

“The retina is one of the most oxygen and energy demanding tissues in the body to generate and transmit visual evoked potential signals, and is directly exposed to visible light and ultraviolet radiation, thus, a large amount of ROS are generated.71 ”

The three types of ultraviolet rays that reach the ground affect the anterior segment of the eye, but are mostly absorbed by the cornea and lens and do not reach the retina.

Numerous in vitro experiments have investigated UV damage to RPE.

However, it is not clear whether this experimental system affects the retinal diseases seen in actual clinical practice.

I'm not sure Ref.71 is the best citation.

Response: Thanks for the referee’ s suggestion. As you said, Ref.71 is not appropriate. We changed the reference to Ref.76-77, and revised the sentence to “The retina is one of the most oxygen and energy demanding tissues in the body to generate and transmit visual evoked potential signals, and is directly exposed to visible light (400-760 nm). Energy metabolism naturally generates oxidative stress, thus, a large amount of ROS is generated.” in the revised manuscript line 227-230.

Reviewer 2 Report

The paper entitled “m6A modification – association with oxidative stress and implications on eye diseases” by Yueqi Ni et al., is a review on potential N6- 15 methyladenosine modification in the development of oxidative stress related eye diseases.

This paper is interesting, however the authors may wish to consider the following prior to publication.

Introduction:

m6A has an important role in maintaining normal dopaminergic function in the brain, since retina is part of brain it is reasonable hypothesize that this impact also on eye diseases (please report the following relevant paper: Pharmacol Ther. 2019 Nov;203:107392. doi: 10.1016/j.pharmthera.2019.07.003). Please add this sentence somewhere on chapter 3.

Chapter 3.5.1 Line 240. Please add the following relevant paper: Invest Ophthalmol Vis Sci. 2009 Aug;50(8):3846-52. doi: 10.1167/iovs.08-3328

The authors claimed that Structure-based virtual screening of compounds is an im- 552 portant approach to obtain effective molecular drugs, they should add that this approach is used in ocular field too (please report the relevant paper Biochem Pharmacol. 2018 Dec;158:13-26. doi: 10.1016/j.bcp.2018.09.016)

Please revise the English style, some typo in the text.

Author Response

The paper entitled “m6A modification – association with oxidative stress and implications on eye diseases” by Yueqi Ni et al., is a review on potential N6- 15 methyladenosine modification in the development of oxidative stress related eye diseases.

This paper is interesting, however the authors may wish to consider the following prior to publication.

m6A has an important role in maintaining normal dopaminergic function in the brain, since retina is part of brain it is reasonable hypothesize that this impact also on eye diseases (please report the following relevant paper: Pharmacol Ther. 2019 Nov;203:107392. doi: 10.1016/j.pharmthera.2019.07.003). Please add this sentence somewhere on chapter 3.

Chapter 3.5.1 Line 240. Please add the following relevant paper: Invest Ophthalmol Vis Sci. 2009 Aug;50(8):3846-52. doi: 10.1167/iovs.08-3328

The authors claimed that Structure-based virtual screening of compounds is an im- 552 portant approach to obtain effective molecular drugs, they should add that this approach is used in ocular field too (please report the relevant paper Biochem Pharmacol. 2018 Dec;158:13-26. doi: 10.1016/j.bcp.2018.09.016)

Please revise the English style, some typo in the text.

Response: Thanks for the referee’ s suggestion. The recommended sentences and references have been inserted in proper places. They are Ref.79, 89 and 138, and in the revised manuscript line 232-234, line 244, and line 424, respectively. We fixed some typo in the text.

Reviewer 3 Report

Section 5 seems unnecessary and extraneous, since the paper is focused on eye diseases, according to the title.

The retina section neglects to discuss age-related macular degeneration. That is an important disease that relates to senescence, which is mentioned in the paper, so it should be addressed here.

Author Response

Comment 1:

Section 5 seems unnecessary and extraneous, since the paper is focused on eye diseases, according to the title.

Response: Thanks for the referee’ s suggestion. Review 1 also pointed out that section 5 was unnecessary. We therefore deleted section 5, figure2, table2, associated abbreviations, and original references 124-152.

Comment 2.

The retina section neglects to discuss age-related macular degeneration. That is an important disease that relates to senescence, which is mentioned in the paper, so it should be addressed here.

Response: Thanks for the referee’ s suggestion. As you said, AMD is an important disease that relates to senescence. We have added description of AMD in revised manuscript line 306-324 (3.5.4), and the added references are Ref 105-109. We also modified the figure 1 and table 1 accordingly.

Reviewer 4 Report

This is an interesting review about the involvement of N6-methyladenosine (m6A) modification  (methylation and demethylation) in oxidative stress in different eye tissues, and in different diseases of the eye. 

Modifications of m6A are implicated in a variety of eye diseases, from cornea, lens to various retinal diseases. Enzymes involved in m6A modification are different methyltransferases (METTL), demethylases (ALKBH5, FTO), and enzymes involved in transcription, processing and translation (e.g. YTHDF1,2). 

Following the description of different eye diseases in which m6A modification is implicated, the next section deals with treatment, i.e. gene therapy and production of antioxidants. 

The last section describes various other diseases related to oxidative stress in which m6A modification is implicated, such as ischemic reperfusion, senescence, chemical damage, , neurodegeneration, diabetes, cancer and others. The conclusion paragraph discusses the applicability of various cancer drug to inhibit m6A regulatory enzymes. These drugs could presumable also be used for eye diseases involving m6A modification. 

General comments: This is a well written review with 2 Figures and 2 tables. 

Specific comment: (Minor) In the reference list, the reference numbers are repeated twice. 

Author Response

Comment: In the reference list, the reference numbers are repeated twice.

Response: Thanks for the referee’ s suggestion. The repeated reference numbers have been modified.

Round 2

Reviewer 1 Report

The manuscript accepted the suggestions and was an improvement over the previous manuscript.

Author Response

thanks 

Reviewer 3 Report

The authors have responded appropriately to reviewer suggestions.

Author Response

thanks 

Reviewer 4 Report

The authors have addressed the reviewer's comments. 

Author Response

thanks